# Absence of transmission of vYF next generation Yellow Fever vaccine in mosquitoes

Rachel Bellone[1], Laurence Mousson[1], Chloé Bohers[1], Nathalie Mantel[2☯], Anna-Bella Failloux [1☯]*

**1** Institut Pasteur, Université Paris Cité, Arboviruses and Insect Vectors, Paris, France, **2** Sanofi—Vaccine Research and Development, Marcy L'Etoile, France

☯ These authors contributed equally to this work.
* anna-bella.failloux@pasteur.fr

**Data Availability Statement:** All relevant data are within the manuscript and its Supporting Information files.

## Abstract

One of the most effective vaccines against an arbovirus is the YFV-17D live-attenuated vaccine developed in 1937 against Yellow Fever (YF). This vaccine replicates poorly in mosquitoes and consequently, is not transmitted by vectors. Vaccine shortages, mainly due to constrained productions based on pathogen-free embryonated eggs, led Sanofi to move towards alternative methods based on a state-of-the-art process using continuous cell line cultures in bioreactor. vYF-247 is a next-generation live-attenuated vaccine candidate based on 17D adapted to grow in serum-free Vero cells. For the development of a new vaccine, WHO recommends to document infectivity and replication in mosquitoes. Here we infected *Aedes aegypti* and *Aedes albopictus* mosquitoes with vYF-247 vaccine compared first to the YF-17D-204 reference Sanofi vaccines (Stamaril and YF-VAX) and a clinical human isolate S-79, provided in a blood meal at a titer of 6.5 Log ffu/mL and secondly, to the clinical isolate only at an increased titer of 7.5 Log ffu/mL. At different days post-infection, virus replication, dissemination and transmission were evaluated by quantifying viral particles in mosquito abdomen, head and thorax or saliva, respectively. Although comparison of vYF-247 to reference vaccines could not be completed to yield significant results, we showed that vYF-247 was not transmitted by both *Aedes* species, either laboratory strains or field-collected populations, compared to clinical strain S-79 at the highest inoculation dose. Combined with the undetectable to low level viremia detected in vaccinees, transmission of the vYF-247 vaccine by mosquitoes is highly unlikely.

## Author summary

Arboviruses such as yellow fever virus (YFV) are transmitted between vertebrate hosts through mosquito bites. Beside vector control, vaccination is a preventive measure to control the spread of the disease. Until now, only the safe and effective YFV-17D live-attenuated vaccine is widely used worldwide. However, its production is fastidious and may not meet global demand in case of large yellow fever outbreaks. A next-generation live-

**Funding:** The project has been supported by Sanofi. The funders had no role in study design, data collection and analysis, decision to publish, or preparation of the manuscript.

**Competing interests:** Nathalie Mantel is an employee of Sanofi and may hold company stocks and/shares. The other authors declare that they have no conflict of interest.

attenuated candidate, vYF-247 grown on serum-free Vero cells, has been developed. Here we studied the infection, dissemination and transmission of vYF-247 vaccine candidate compared primarily to an YF clinical isolate S-79 and secondarily to two reference Sanofi vaccines (Stamaril and YF-VAX) in mosquitoes, *Aedes aegypti* and *Aedes albopictus*. Although no significant conclusions could be drawn from the comparisons to the reference vaccines, we nevertheless showed that vYF-247 was unable to be transmitted by mosquitoes at the highest inoculation dose. Our study corroborates that vYF-247 adapted to growth on Vero cells has maintained YFV-17D inability to be transmitted by mosquitoes, which remains a key feature to develop a live attenuated vaccine that will be distributed to millions of subjects through vaccination campaigns.

## Introduction

Yellow fever (YF) is caused by a flavivirus belonging to the family *Flaviviridae* and is transmitted by the *Aedes aegypti* mosquito. Most likely originating in Africa, it was introduced into the Americas in the 1600s during the slave trade, alongside the mosquito vector [1]. It was the beginning of devastating outbreaks, mainly in port cities where most people were immunologically naïve and susceptible to infection. The description in 1886 by the Cuban physician Carlos Finlay of transmission of YF causative agent by the mosquito *Ae. aegypti*, confirmed 20 years later by Walter Reed, marked a turning point in the control of YF [2,3]. Mosquito eradication campaigns of the 1940s and 1950s led to the control of YF in cities of the Americas and the Caribbean [4]. However, the relaxation of control measures led to the land recolonization by *Ae. aegypti* [5] and later, to the introduction and establishment of *Aedes albopictus* [6].

YF global burden is still high with the disease endemic in tropical and subtropical regions of South America and Africa [1]. The virus circulates originally within a sylvatic cycle between non-human primates and canopy-dwelling mosquitoes. Spillovers are detected in areas termed zones of emergence, described as a savannah cycle mainly in Africa. Urban outbreaks are occasionally reported mainly in regions with *Ae. aegypti* as the main vector [7]. While urban transmission of yellow fever virus (YFV) can be controlled by public health interventions, the sylvatic cycle is much harder to control given the diversity of wild habitats. Recently, urban outbreaks of YF have been reported in several states in Nigeria [8] and Brazil [9]. The YF incidence is estimated to be between 200,000 and 300,000 cases per year. Travelers from Angola brought 11 cases to China threatening YF-free countries with a new epidemic [10].

Following the isolation of YFV (strain Asibi) in 1927, the YFV-17D vaccine strain was developed by passages of Asibi through chicken and mouse tissues, this vaccine being used today throughout the world. The YFV-17D vaccine is a safe and low-cost vaccine that confers long-duration immunity [11]. YFV-17D can infect *Ae. aegypti* midgut [12] but does not disseminate to other internal tissues and thus is not excreted in mosquito saliva. Increased demand for YF immunization (routine immunization, prevention campaigns and outbreak response) over the last decade has led to an increased risk of global YF vaccine shortages in case of large outbreaks [13]. These shortages have been partly worsened by insufficient availability of specific pathogen-free embryonated eggs required for timely vaccine production.

A new live-attenuated YF vaccine candidate (referred to as vYF-247) cloned from a YFV-17D vaccine (YF-VAX) sub-strain adapted for growth in Vero cells cultured in serum-free media is currently in development. The vYF-247 vaccine candidate is reported to be safe, immunogenic and able to induce protection from lethal challenge in small animal models (mice and hamsters) [14]. Here, we tested the replication of the vaccine candidate vYF-247 in

mosquitoes, *Ae. aegypti* and *Ae. albopictus* in comparison with the clinical isolate S-79 but also the YF-17D-204 reference vaccines (Sanofi Stamaril and YF-VAX vaccine drug substance amplified by one passage on Vero cells) when feasible.

## Methods

### Ethic statements

Animals were housed in the Institut Pasteur animal facilities (Paris) accredited by the French Ministry of Agriculture for performing experiments on live rodents. Work on animals was performed in compliance with French and European regulations on care and protection of laboratory animals (EC Directive 2010/63, French Law 2013–118, February 6th, 2013). All experiments were approved by the animal experimentation ethics committee #89 and registered under the reference APAFIS#6573-201606l412077987 v2.

### Mosquito populations

Mosquitoes (Table 1) were reared in an insectary at the Institut Pasteur in controlled conditions (24±1˚C, 70% relative humidity, a 12:12 hour (Light:Dark) photoperiod). Larvae were distributed in pans (200 larvae/pan) containing 1.5 L of dechlorinated tap water supplemented with yeast tablets. Obtained adults were placed in cages and daily provided with 10% sucrose solution until infection.

### Viral strains

Four viral strains were used: the vaccine candidate vYF-247 [14], the two live-attenuated YF-17D-204 reference vaccines, Sanofi Stamaril and YF-VAX, and a clinical isolate S-79 (accession number: MK060080, [15]). vYF-247 (batch #FDV02926) was produced at large scale in bioreactor on serum-free Vero cells in Sanofi proprietary medium that does not contain antibiotics or any product from animal or human origin. Stamaril and YF-VAX batches were amplified once on Vero cells from egg-based vaccine bulks produced by Sanofi and sent to Institut Pasteur. Vero cells were cultured in Iscove's Modified Dulbecco's Medium (IMDM)(Thermo Fisher) supplemented with 4% Fetal Calf Serum (FCS) and infected at MOI 0.001 in Hyperflask (Corning, New York USA). Supernatants were collected at 7 days post-infection and concentrated 5-fold on Centricon Plus-70, 100K device (Millipore, Massachussets USA). S-79 was isolated from a patient returning from Senegal in 1979, passaged twice on mice brains and twice on C6/36 cells [15].

### Mosquito experimental infections

Boxes of 60 one-week-old female adults were fed for 15 min through a pig intestine membrane covering the base of a feeder (Hemotek membrane feeding system, UK) containing 1.4 mL of

**Table 1. List of mosquito colonies/populations used for YFV infections.**

| Mosquito species | Mosquito strain | Origin | Date of collection | Generation used for experimental infections |
|---|---|---|---|---|
| *Aedes aegypti* | Paea | French Polynesia | 1994 | Lab colony |
| | Les Abymes | Guadeloupe | 2020 | F2 |
| | Taiwan Mid West | Taiwan | 2019 | F4 |
| | Yaoundé | Cameroon | 2020 | F4 |
| | Villa Yolanda | Colombia | 2020 | F4 |
| *Aedes albopictus* | La Providence | La Réunion | 2007 | Lab colony |
| | Tainan | Taiwan | 2019 | F5 |

rabbit erythrocytes supplemented with 10 mM adenosine triphosphate (ATP) as a phagosti-mulant and 0.7 mL of viral stock to obtain a final titer of $10^{6.5}$ and $10^{7.5}$ ffu/mL. Only fully engorged mosquitoes were kept and maintained in containers placed in climatic chambers at 28˚C ± 0·1˚C until processing at different days post-infection (dpi). Mosquitoes were fed with 10% sucrose solution.

### Analysis of vector competence

Batches of 20–40 females were analysed at 7-, 14- and 21-days post-infection (dpi) for lab colonies, and only 21 dpi for field-collected mosquitoes. After cold anaesthesia, wings and legs of each mosquito were removed and the proboscis was inserted into 20 µL tip filled with Fetal Bovine Serum for saliva collection [16]. Abdomen and head+thorax (HT) were separated from each mosquito and ground individually in 300 µL of L15 medium supplemented with 2% of FBS. Homogenates (abdomen and HT) and saliva were titrated by focus fluorescent assay on *Ae. albopictus* C6/36 cells. After 10-fold dilutions, samples were inoculated onto C6/36 cells in 96-well plates. After a 5-day incubation period at 28˚C, cells were fixed with 3.6% formaldehyde, washed and hybridized with YFV specific primary antibody OG5 NB100-64510 (Novusbio, CO, USA) and revealed by using a fluorescent-conjugated secondary antibody, Alexa Fluor 488 goat anti-mouse IgG (Life Technologies, CA, USA). Titers are in focus forming units (ffu).

Three parameters were used to describe the viral infection, dissemination and transmission. Infection rate (IR) corresponds to the proportion of mosquitoes with an infected abdomen among tested mosquitoes. Dissemination efficiency (DE) refers to the proportion of mosquitoes with infected head and thorax (HT) among tested mosquitoes. Transmission efficiency (TE) is the proportion of mosquitoes with infectious saliva among tested mosquitoes.

### Statistical analysis

Statistical tests were conducted using the STATA software (StataCorp LP, Texas, USA) and R 4.0.3. P-values below 0.05 were considered significant.

## Results

### Replication of vYF-247 in two laboratory colonies

We measured the susceptibility of two lab colonies, *Ae. aegypti* Paea (AAPAEA) and *Ae. albopictus* La Providence (ALPROV) to vYF-247 compared to the YF-17D-204 reference Sanofi vaccines (Stamaril and YF-VAX) and the clinical isolate S-79. First, we scored the infection, dissemination and transmission status of 953 mosquitoes at 7, 14, and 21 dpi with a blood meal at a titer of $10^{6.5}$ ffu/mL. When exposing AAPAEA to all four viral strains, infection was only observed with YF-VAX (2.5%; 1/40) and vYF-247 (2.5%; 1/40) at 14 dpi, and Stamaril (2.77%; 1/36) at 21 dpi. We detected dissemination only at 14 dpi with vYF-247 and no transmission whatever the viral strain used (S1 Fig). ALPROV was able to become infected at 7 dpi with Stamaril (20%; 8/40); at 14 dpi with Stamaril (7.5%; 3/40), YF-VAX (7.5%; 3/40) and vYF-247 (5%; 2/40); and at 21 dpi with Stamaril (7.5%; 3/40), YF-VAX (7.5%; 3/40) and S-79 (2.5%; 1/40). We observed dissemination only at 7 dpi with Stamaril (2.5%; 1/40) and no transmission whatever the viral strain (S2 Fig).

Then we decided to increase the blood meal titer to $10^{7.5}$ ffu/mL but we only succeeded in getting high titers of viral productions for vYF-247 and S-79. Therefore, these two viral strains were the only ones tested in the following experiments. We examined the infection, dissemination and transmission status of 398 mosquitoes at 7, 14, and 21 dpi with a blood meal at a titer of $10^{7.5}$ ffu/mL.

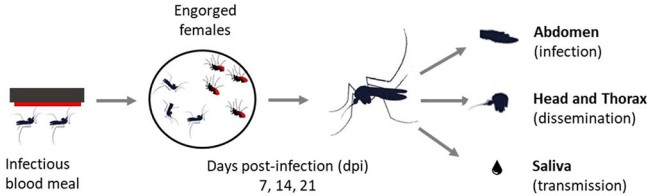

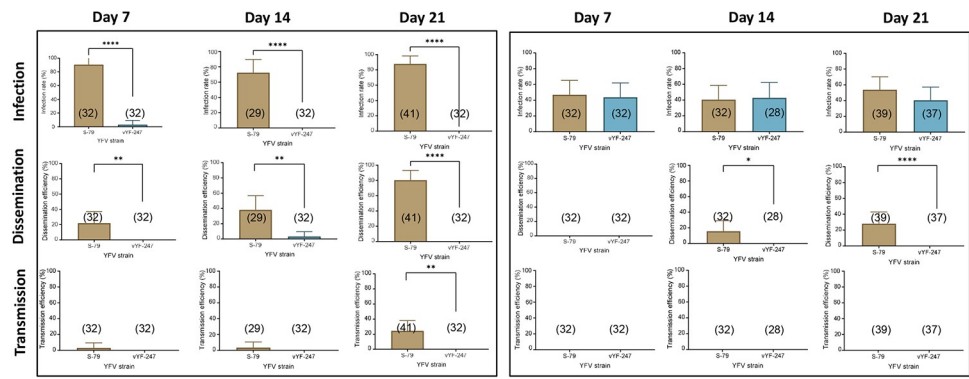

**Fig 1. Infection rate, dissemination efficiency, and transmission efficiency of *Aedes aegypti* AAPAEA and *Aedes albopictus* ALPROV exposed to vYF-247 and S-79 strains.** Mosquitoes were presented to a blood meal provided at a titer of $10^{7.5}$ ffu/mL and examined at 7, 14, and 21 days post-infection. In brackets, number of tested mosquitoes. Stars indicate statistical significance of comparisons by Fisher's exact test: $^{*}p \leq 0.05$, $^{**}p \leq 0.01$, $^{****}p \leq 0.0001$.

When exposing AAPAEA to vYF-247, infection was only detected at 7 dpi (3.12%; 1/32), dissemination at 14 dpi (3.12%; 1/32), and no transmission whatever the dpi (Fig 1). With the YFV S-79, we detected infection (IR: 90.6% (29/32) at 7 dpi; 72.4% (21/29) at 14 dpi; and 87.8% (36/41) at 21 dpi), dissemination (DE: 21.8% (7/32) at 7 dpi; 37.9% (11/29) at 14 dpi; and 80.4% (33/41) at 21 dpi), and transmission (TE: 3.1% (1/32) at 7 dpi; 3.4% (1/29) at 14 dpi; and 24.4% (10/41) at 21 dpi) in AAPAEA (Fig 1).

With ALPROV exposed to vYF-247, there were high IRs at all three dpi (43.75% (14/32) at 7 dpi; 42.85% (12/28) at 14 dpi; and 40.54% (15/37) at 21 dpi), and no dissemination nor transmission (Fig 1). When exposing to S-79, infection and dissemination were detected (IR: 46.8% (15/32) at 7 dpi; 40.6% (13/32) at 14 dpi; and 53.8% (21/39) at 21 dpi), dissemination (DE: 0% (0/32) at 7 dpi; 15.6% (5/32) at 14 dpi; and 28.2% (11/39) at 21 dpi) but no transmission was observed in ALPROV (Fig 1).

We confirmed that the vaccine candidate vYF-247 was not transmitted by *Ae. aegypti* compared to the YFV S-79 (Fisher's exact test: p < 0.01) and neither vYF-247 nor YFV- S-79 were transmitted by *Ae. albopictus*.

To compare the number of viral particles produced by mosquitoes, we titrated abdomen, HT and saliva from each mosquito. In AAPAEA infected with vYF-247, we found $10^{2.3}$ viral particles at 7 dpi in a single abdomen, $10^{1.5}$ viral particles at 14 dpi in one HT and no virus detected in saliva (Fig 2). With S-79, the mean numbers of viral particles were high in AAPAEA: $10^{3.2}$ at 7 dpi, $10^{3.5}$ at 14 dpi, and $10^{3.3}$ at 21 dpi in abdomens; $10^{2.6}$ at 7 dpi, $10^{5.0}$ at 14 dpi, and $10^{5.5}$ at 21 dpi in HT; and at 21 dpi, an average of $10^{1.8}$ particles was detected in saliva (N = 10) (Fig 2).

In ALPROV infected with vYF-247, viral particles were only detected in abdomens: mean number of $10^{2.2}$ viral particles at 7 dpi, $10^{2.2}$ at 14 dpi and $10^{1.9}$ at 21 dpi (Fig 2). S-79 replicated

**Fig 2. Number of viral particles (Log$_{10}$) in abdomen, HT, and saliva of *Aedes aegypti* AAPAEA and *Aedes albopictus* ALPROV exposed to vYF-247 and S-79 strains.** Mosquitoes were presented to a blood meal provided at a titer of $10^{7.5}$ ffu/mL and examined at 7, 14, and 21 days post-infection. In brackets, number of YFV-positive mosquitoes. Stars indicate statistical significance of comparisons by Kruskal-Wallis test: **p $\leq$ 0.01, ***p $\leq$ 0.001, ****p $\leq$ 0.0001.

well in abdomen ($10^{2.8}$ at 7 dpi, $10^{3.8}$ at 14 dpi, and $10^{3.9}$ at 21 dpi) and HT (0 at 7 dpi, $10^{3.8}$ at 14 dpi, and $10^{4.3}$ at 21 dpi) but no virus was detected in saliva of ALPROV (Fig 2).

We showed that when vYF-247 succeeded in infecting mosquitoes, viral replication was less important (see Fig 2, abdomen in *Ae. albopictus*) compared to the YFV S-79 (Mann-Whitney test: p < 0.05).

## Replication of vYF-247 in field-collected populations

To expand our assessment of vYF-247, we examined a panel of five field-collected populations of *Ae. aegypti* [4] and *Ae. albopictus* [1] originating from Guadeloupe, Cameroon, Colombia, and Taiwan. We measured the susceptibility of 361 mosquitoes to vYF-247 compared to the YFV S-79 using the same standardized artificial feeding protocol and a single measure at 21 dpi.

When exposing the four *Ae. aegypti* populations to vYF-247, infection was only detected for Les Abymes (IR = 7.5%; 3/40) and Villa Yolanda (IR = 23.08%; 3/13), and infection plus dissemination for Taiwan Mid-West (IR = 10%; 4/40; DE = 5%; 2/40). No transmission of vYF-247 was detected (Fig 3). With S-79, we detected high IRs for Les Abymes (100%; 33/33), Taiwan Mid-West (90%; 36/40), and Villa Yolanda (97.2%; 35/36), and a low IR of 17.5% (7/40) for Yaoundé (Fisher's exact test: p < 0.05). DEs varied from 7.5% (Yaoundé, 3/40) to 81.81% (Les Abymes, 27/33) (Fisher's exact test: p < 0.05) and TEs from 5% (Yaoundé, 2/40) to 33.33% (Les Abymes, 11/33) (Fisher's exact test: p < 0.05) (Fig 3). We showed that the four *Ae. aegypti* populations successfully transmitted the YFV S-79 while they were refractory to transmit vYF-247.

When examining viral replication in mosquitoes, the number of viral particles in abdomen, HT and saliva was low when infected with vYF-247 (abdomen: from $10^{1.3}$ ffu (Les Abymes) to $10^{2.6}$ ffu (Villa Yolanda) and HT: $10^{2.2}$ ffu (Taiwan Mid-West)) compared to mosquitoes infected with S-79 (abdomen: from $10^{2.9}$ ffu (Yaoundé) to $10^{4.0}$ ffu (Taiwan Mid-West), HT: from $10^{4.0}$ ffu (Villa Yolanda) to $10^{4.4}$ ffu (Taiwan Mid-West), saliva: from $10^{2.1}$ ffu (Villa Yolanda) to $10^{2.9}$ ffu (Yaoundé)) (Fig 3). Except saliva, the numbers of viral particles in abdomen and HT were significantly different according to population (Mann-Whitney test: p < 0.05) (Fig 3).

In addition to the four populations of *Ae. aegypti*, we also infected one population of *Ae. albopictus*, Tainan. With vYF-247, we observed a viral infection (IR: 12.8%) but no dissemination or transmission. The mean number of viral particles in abdomen was $10^{1.8}$ (Fig 4). However, with S-79, a successful viral infection (IR: 32.5%), dissemination (DE: 20%) and transmission (TE: 7.5%) were detected. The mean number of viral particles in abdomen, HT and saliva were very high: $10^{3.6}$ in abdomen, $10^{4.5}$ in HT, and $10^{3.4}$ in saliva (Fig 4).

## Discussion

Altogether, our results indicate that the vaccine candidate vYF-247 was not transmitted by *Ae. aegypti* and *Ae. albopictus* populations tested when provided in an artificial blood meal.

Examples of insufficient vaccination coverage include the YF outbreak in Angola in 2016, the imported cases of YF into China due to unvaccinated travelers returning from endemic areas, and the detection of YF cases near Rio de Janeiro, city which had been free of YF since 1954. WHO advocates that at least 80% of vaccine coverage would be necessary to prevent and control such outbreaks [17]. The present annual production of YF vaccine from all manufacturers globally is estimated to be around 80 million doses per year, and only 5–6 million doses can be mobilized immediately [18]. Even if the global supply has increased in the past years and been secured through the Eliminate Yellow fever Epidemics (EYE) program, the risk of

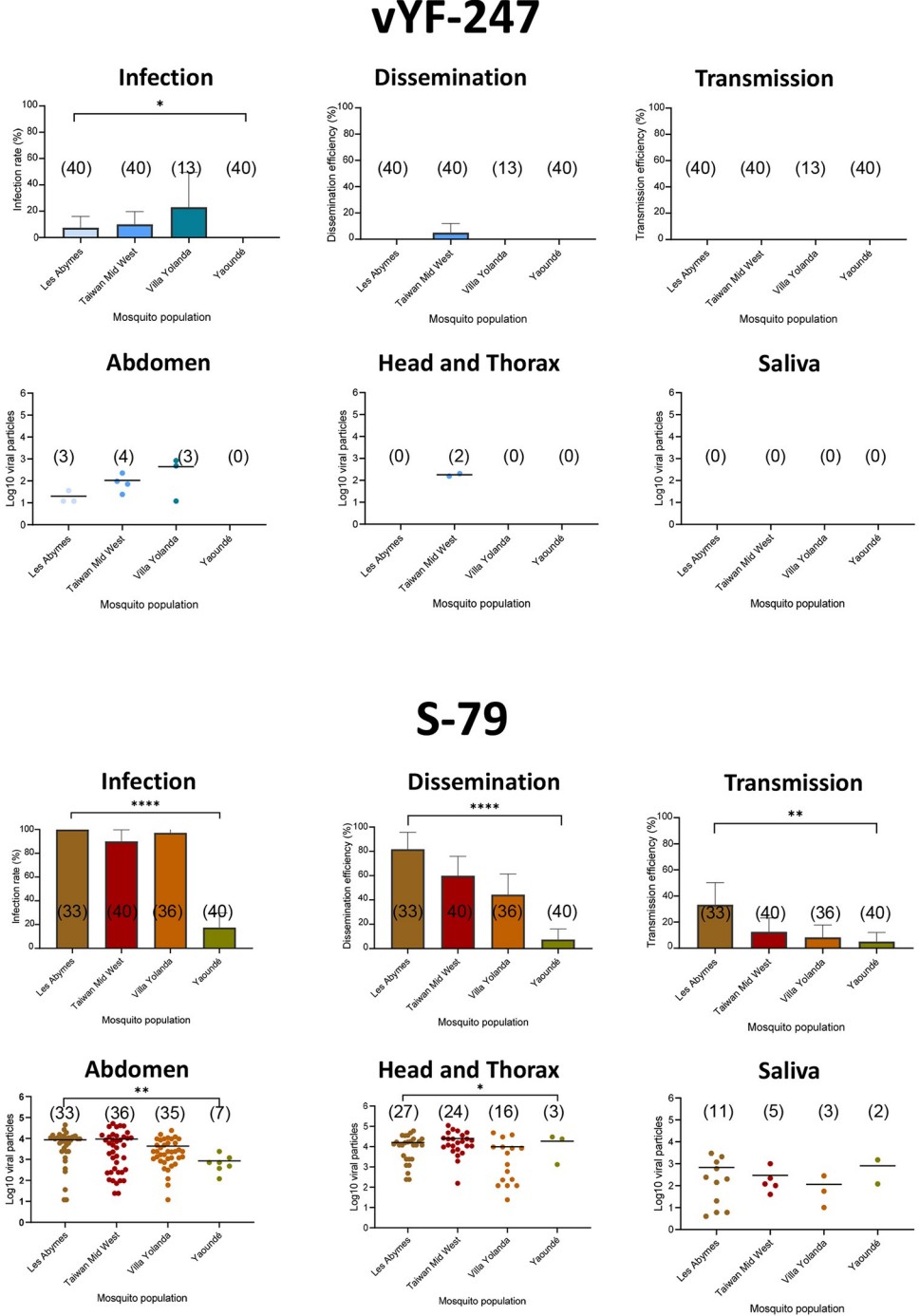

**Fig 3. Infection rate, dissemination efficiency, transmission efficiency, Log$_{10}$ viral particles in abdomen, HT, and saliva of 4 *Aedes aegypti* populations exposed to vYF-247 and YFV S-79 strains.** Mosquitoes were presented to a blood meal provided at a titer of $10^{7.5}$ ffu/mL and examined at 21 days post-infection. In brackets, number of tested mosquitoes for IR, DE, TE, and number of YFV-positive abdomens, HT and saliva. Stars indicate statistical significance of comparisons: *p ≤ 0.05, **p ≤ 0.01, ****p ≤ 0.0001.

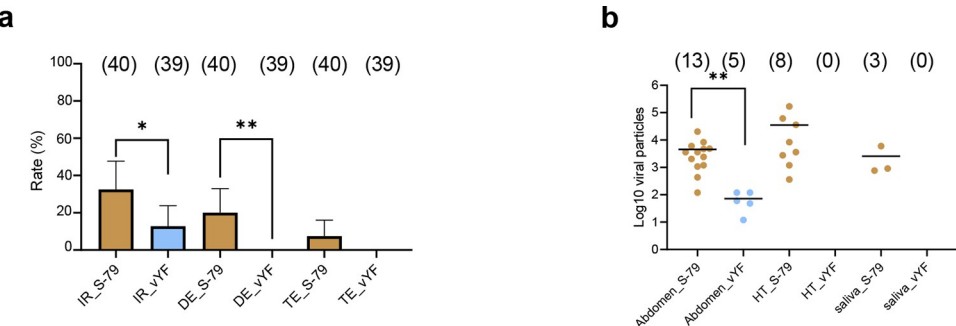

**Fig 4. Infection rate, dissemination efficiency, transmission efficiency (a), Log$_{10}$ viral particles in abdomen, head and thorax, and saliva, and saliva (b) of *Aedes albopictus* Tainan (Taiwan) exposed to vYF-247 and S-79 strains.** Mosquitoes were presented to a blood meal at a titer of $10^{7.5}$ ffu/mL and examined at 21 days post-infection. In brackets, number of tested mosquitoes for IR, DE, TE, and number of YFV-positive abdomens, HT and saliva. Stars indicate statistical significance of comparisons: $^{*}$p $\leq$ 0.05, $^{**}$p $\leq$ 0.01.

shortage in case of large outbreaks stays high. The live-attenuated YF-17D vaccine, available since 1937, is safe and confers life-long immunity [19]. A major blocking-point is that the manufacturing process requires embryonated chicken eggs [20]. A new live-attenuated YF vaccine candidate selected and cloned from a YF-17D vaccine sub-strain adapted for growth on Vero cells (referred to as vYF) cultured in serum-free medium has been developed and is under clinical study in healthy volunteers [14].

At the outset of this study, we performed inoculations of two laboratory colonies of mosquitos with four viral strains of YF provided at $10^{6.5}$ ffu/mL but both colonies were poorly infected and failed to disseminate the virus. When deciding to increase viral titers in blood meals, we were not able to generate viral suspensions at a titer ~ $10^{7.5}$ ffu/mL for either Stamaril or YF-VAX, both being issued from embryonated chicken eggs compared to vYF-247 and S-79 produced in cell culture, and therefore proceeded with vYF-247 & S-79 only. The impossibility to compare head-to-head the replication in mosquitoes of the vYF-247 vaccine candidate to the reference vaccine at the highest inoculation dose is the main limitation of our study. Indeed, results obtained from the comparison of the vYF-247 vaccine candidate to the reference vaccine at $10^{6.5}$ ffu/mL did not yield interpretable data since clinical strain S-79 was not either transmitted at this titer. The WHO-TRS 978-Annex 5: *Recommendations to assure the quality, safety and efficacy of live attenuated yellow fever vaccines* recommends to assess mosquito infectivity and dissemination of any new YF vaccine candidates in comparison with a currently acceptable vaccine [21]. Thus, further comparisons of vYF-247 vaccine candidate to the reference vaccines will be required to complete this work.

When infecting with a blood meal at $10^{7.5}$ ffu/mL, we show that vYF-247 cannot be transmitted by two laboratory colonies and five field-collected populations of *Ae. aegypti* and *Ae. albopictus*. The upper limit of viremia detected in blood of A129 mice 4 days post-inoculation with vYF-247 is approximately 7 LogGeq/mL [14]. Of note, these mice are deficient for type I interferon receptors required to initiate innate and adaptive immune responses involved in viral clearance and so are highly permissive for viral replication. The viremia generated after vYF-247 vaccination in non-human primates or humans are very low (below 6 LogGeq/mL) and transient (detected for maximum 3 days). Therefore, the low viremia induced after vaccination combined with the mosquitoes' inability to transmit vYF-247 indicates that transmission of vYF-247 through a mosquito bite is highly unlikely.

It has been shown that the YFV-17D vaccine infects the midgut but does not spread to secondary organs in the mosquito vector [12]. YFV-17D vaccine differs from the clinical strain

YFV-Asibi by 12 mutations in the envelope protein. Non-synonymous mutations were located in the domain III offering different affinities for a cell entry receptor [22]. It is likely that the receptor used by YFV-17D is poorly expressed at the apical surface of mosquito midgut epithelial cells. YFV-17D vaccine also generates viral populations with low diversity as compared to its parental strain Asibi [23]. This low diversity may limit viral dissemination in mosquitoes [24]. As vYF-247 derives from YFV-17D, it is probable that both vaccines share some properties [25].

It is estimated that more than half of total global population lives in countries where *Ae. aegypti* and *Ae. albopictus* proliferate [26]. Both species meet all criteria to trigger outbreaks: human-biting mosquitoes, high densities, good survival rate, and susceptibility to YFV [27]. Part of the failure to control arboviral diseases is due to ineffective vector control, itself a consequence of insecticide resistance in mosquito populations [28] and presence of cryptic breeding sites which are difficult to implement vector control [29]. The increased number of YF-infected travelers despite the International Health Regulations, poor control of YF vaccination at entry to some countries and falsified vaccine certificates, increase the risk of introducing YF into new areas [30,31]. Thus, the risk of transmission of YF in naïve populations and large-scale outbreaks of YF is not a question of "if" but "when". To avoid future devastating outbreaks similar or worse than previous outbreaks, and given the high fatality rate, reaching 86% in Nigeria [32], an emergency plan must be carefully prepared.

In addition to ensuring adequate supplies of diagnostic tests, insecticides and repellants, and antivirals, improved control at borders, and readily available medical care, there must be an adequate supply of YF vaccine that can be rapidly distributed to outbreak areas. This should be achievable with vYF. vYF-247 is currently being evaluated in phase II clinical trials and may be an alternative to the egg-based manufactured vaccine, while maintaining the attenuation and efficacy features of currently approved vaccines but without the manufacturing technical issues. Moreover, the absence of transmission by the two *Aedes* species combined with the low viremia developed by vaccinated people support that vYF-247 is a good vaccine candidate.

## Supporting information

**S1 Fig. Infection rate (IR), dissemination efficiency (DE), and transmission efficiency (TE) of *Aedes aegypti* Paea exposed to 4 YFV strains (Stamaril, YF-VAX, vYF-247, and S-79) provided in a blood meal at $10^{6.5}$ FFU/mL and examined at 7, 14, and 21 days post-infection.** In brackets, number of mosquitoes. IR corresponds to the proportion of mosquitoes with an infected abdomen among tested mosquitoes. DE refers to the proportion of mosquitoes with infected HT among tested mosquitoes. TE is the proportion of mosquitoes with infectious saliva among tested mosquitoes.
(TIF)

**S2 Fig. Infection rate (IR), dissemination efficiency (DE), and transmission efficiency (TE) of *Aedes albopictus* La Providence exposed to 4 YFV strains (Stamaril, YF-VAX, vYF-247, and S-79) provided in a blood meal at $10^{6.5}$ FFU/mL and examined at 7, 14, and 21 days post-infection.** In brackets, number of mosquitoes. IR corresponds to the proportion of mosquitoes with an infected abdomen among tested mosquitoes. DE refers to the proportion of mosquitoes with infected HT among tested mosquitoes. TE is the proportion of mosquitoes with infectious saliva among tested mosquitoes.
(TIF)

## Acknowledgments

The authors thanks Manuel Vangelisti, Sophia Mundle and Bachra Rokbi for the discussions and critical review of the manuscript.

## Author Contributions

**Conceptualization:** Nathalie Mantel, Anna-Bella Failloux.

**Formal analysis:** Rachel Bellone, Laurence Mousson, Chloé Bohers.

**Funding acquisition:** Nathalie Mantel, Anna-Bella Failloux.

**Investigation:** Rachel Bellone, Laurence Mousson, Chloé Bohers.

**Writing – original draft:** Anna-Bella Failloux.

**Writing – review & editing:** Rachel Bellone, Nathalie Mantel, Anna-Bella Failloux.

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
