## [Decision Letter · Decision Letter 0]

20 Aug 2022

Dear Dr Failloux,

Thank you very much for submitting your manuscript "Absence of transmission of vYF next generation Yellow Fever Vaccine in mosquitoes" for consideration at PLOS Neglected Tropical Diseases. As with all papers reviewed by the journal, your manuscript was reviewed by members of the editorial board and by several independent reviewers. In light of the reviews (below this email), we would like to invite the resubmission of a significantly-revised version that takes into account the reviewers' comments. 

We cannot make any decision about publication until we have seen the revised manuscript and your response to the reviewers' comments. Your revised manuscript is also likely to be sent to reviewers for further evaluation.

Sincerely,

Abdallah Samy

Section Editor

Reviewer comments: 

Reviewer's Responses to Questions

**Key Review Criteria Required for Acceptance?**

**Methods**

-Are the objectives of the study clearly articulated with a clear testable hypothesis stated?

-Is the study design appropriate to address the stated objectives?

-Is the population clearly described and appropriate for the hypothesis being tested?

-Is the sample size sufficient to ensure adequate power to address the hypothesis being tested?

-Were correct statistical analysis used to support conclusions?

-Are there concerns about ethical or regulatory requirements being met?

Reviewer #1: The objectives of the present study are clearly articulated with a clear testable hypothesis stated. The study design is apropriated to adress the stated objectives.

Reviewer #2: The 17D live attenuated yellow fever vaccine is grown in pathogen-free embryonated eggs. vYF-247 is a vaccine candidate for yellow fever that is produced from 17D adapted to grow in serum-free Vero cells. The authors aim to study vYF-247 infectivity and replication in mosquitoes. They hypothesize that the vaccine candidate will not be transmissible by mosquitoes, especially when compared to a wild type strain of yellow fever virus (YFV). These objectives and hypothesis are clearly stated.

The study design includes testing vYF, two licensed YF 17D-204 vaccines, and wild type YFV in multiple species of mosquito (lab colonies and field-collected). The mosquito experimental techniques and interpretation of results pertaining to infection, dissemination and transmission are in line with the literature. The study design is appropriate for the stated objectives. However, the paper states, “The WHO-TRS 978-Annex 5: Recommendations to assure the quality, safety and efficacy of live attenuated yellow fever vaccines recommends to assess mosquito infectivity and dissemination of any new YF vaccine candidates in comparison with a currently acceptable vaccine. Here, we tested the replication of the vaccine candidate vYF- 247 in mosquitoes, Ae. aegypti and Ae. albopictus in comparison with the YF-17D-204 reference vaccines (Sanofi Stamaril and YF-VAX).” The study design for the majority of the paper only includes vYF and wild type YF, not the vaccine references. The part of the study design that includes the vaccine references does not have strong data and doesn’t imply a correlation between vYF-247 and the vaccines, between vYF and the wild type, or between the wild type and the reference vaccines. Without any of these correlations, the vYF vaccine candidate cannot be compared to the vaccine references as the authors state in the quote I mentioned above.

The authors should consider doing these experiments in Haemagogus mosquitoes as well, as these mosquitoes can transmit YF in South America. Otherwise, the populations are clearly described and are appropriate for the hypothesis, which is to assess transmissibility from mosquitoes of vYF-247. Again, vYF-247 is not compared to the 17D reference vaccines, therefore, any hypothesis regarding comparison between these is not tested by these experiments.

The sample size for the conclusions made is sufficient.

The explanation of the terminology on line 151-154 is very helpful.

The statistical tests utilized (Fisher’s exact test, Mann-Whitney, Kruskal-Wallis) are used correctly.

There are no concerns regarding ethical or regulatory requirements.

There is a concern for future regulatory requirements. This paper alone cannot be used to justify the approval of the vYF-247 vaccine based on mosquito transmissibility. According to WHO requirements, mosquito transmission of any YF vaccine candidate must be compared to current 17D vaccines, which this paper does not adequately cover.

**Results**

-Does the analysis presented match the analysis plan?

-Are the results clearly and completely presented?

-Are the figures (Tables, Images) of sufficient quality for clarity?

Reviewer #1: The results analysis presented match the analysis plan and the results are clearly and completely presented.

Reviewer #2: Everything in the analysis presented matches the analysis plan except for the comparison between vYF-247 and the vaccine references. In the abstract, the authors state that vYF-247 will be compared to 17D-204 reference vaccines and wild type virus. As the bulk of this paper discusses only a comparison between vYF-247 and the wild type, that statement is not accurate. The wording should be made clearer earlier in the paper (especially in the abstract and introduction) that the results only really compare the vaccine candidate to the wild type, and that there are very limited comparisons to the vaccine strains.

In the experiments that include the YF vaccine references, no real correlation can be seen between vYF-247 and the vaccines, between vYF and the wild type, or between the wild type and the reference vaccines. With this in mind, it is not possible to draw conclusions from the vaccine references (data in the supplemental figures) to the vaccine candidate (data in the main paper).

The authors analyze each mosquito section and mosquito population as intended. They also compare each population tested as intended.

The figures are clear and concise and show all relevant information. The error bars, lines indicating averages, and significance bars are clear. None of the figures are cluttered. The mosquito diagram in figure 1 is particularly helpful to understand the paper. The X and Y axis labels may be a little small but can be easily read on a digital copy. It is easy to distinguish between the colors in each figure.

The results are presented clearly and linearly. They also appear to be completely presented and tell a good story from start to finish.

**Conclusions**

-Are the conclusions supported by the data presented?

-Are the limitations of analysis clearly described?

-Do the authors discuss how these data can be helpful to advance our understanding of the topic under study?

-Is public health relevance addressed?

Reviewer #1: The conclusions are supported by the data presented, the limitations of this study are clearly described and the authors discuss the obtained data to advance our understanding on vYF next generation YF vaccine transmission in mosquitoes. This study is relevant for public health since is recommended by WHO for the development of a new vaccine.

Reviewer #2: The first sentence of the discussion is the main conclusion of this paper and it is supported by the data presented. The conclusions are not overstated and the authors do a good job not to mention comparison to reference vaccines in their statement of the main conclusions. The data collected are easy to understand and the conclusion drawn is clear.

The major limitation of this paper (not being able to compare vYF-247 to the vaccine references) is stated and explained well in the discussion. Their reasoning is understandable and all too relatable. However, this limitation is stated less clearly in the body of the paper and the abstract. The body and abstract should be changed to state that vYF-247 was only successfully compared to the wild type. Any mention of comparison between vYF-247 and the vaccine references should clearly state that this was not fully tested in this paper. Non-inferiority studies are important for the development of next-generation YF vaccines and these studies in mosquitoes should be done with the reference vaccines.

While not necessarily a limitation of this paper, it would be interesting to discuss transmission studies from mosquitoes to mammals with the viral strains used in this paper as a future direction for research.

The discussion of viremia in mice and NHPs following inoculation with vYF-247 is extremely relevant for the progression of vYF-247 through future vaccine approval. These data in combination with the mosquito data presented in this paper build a good case that it is very unlikely that vYF-247 can be transmitted by mosquitoes. 

There are many good references to the current literature on YF vaccines from line 299-316. This helps compare vYF-247 to the current YF vaccines and provides good reasoning for the use of vYF-247.

The public health relevance is discussed well. It is good that they are addressing the public health need for YF vaccination as well as the reasoning for developing next-generation YF vaccines. They present real-world goals such as the WHO EYE initiative and use these goals to advocate for their vaccine candidate.

The last paragraph discusses how use of next-generation YF vaccines can be combined with other infection control measures and is a good addition to this paper.

The gap in knowledge that this analysis fills is clear: can this vaccine candidate be transmitted by mosquitoes? It is a relevant and necessary study that will further our understanding of yellow fever vaccines.

**Editorial and Data Presentation Modifications?**

Reviewer #1: In the article “Absence of transmission of vYF next generation Yellow Fever Vaccine in mosquitoes”, Bellone and colleagues describe the results obtained by vector competence studies of Ae. aegypti and Ae. albopictus mosquitoes for vFY-247 vaccine compared to YF-17D-204 reference vaccines and a human isolate. This work corresponds to the first-time experimental infection in mosquitoes for vYF next generation yellow fever vaccine. The obtained results are relevant to develop new life attenuated vaccines following WHO’s recommendations. 

My review includes a few minor suggestions (text corrections or comments requested to the authors).

Lines 30: Please, change midgut by abdomen since the midgut was not dissected.

Lines 31: Please, eliminate “did not disseminate” since the authors observed dissemination (1/32) when Ae. aegypti colony (AAPAEA) and field-collected population from Taiwan Mid-West population (2/40) were exposed to vYF-247.

Line 66: Please, change the reference 1 by other reference more recent. 

Lines 66-69: Please, add a reference. 

Lines 79-80: Do all the references 11-13 fit well with the sentence: YFV-17D can infect Ae. aegypti midgut? Which(s) reference(s) support that YFV-17D does not disseminate to other internal tissues? In the discussion section (lines 308-309), it is related with reference 12, but could reference 13 be adequate? Please, add the adequate reference at the end of this sentence.

Line 93: Please, add the reference of this Annex 5. 

Lines 93-95: Please, give more details on YF-17-204 and reference vaccines origins.

Line 109: Why do the authors rear the mosquitoes at 24ºC and 12:12 hour photoperiod. 

Line 112: When, in terms of age, did the mosquitoes were exposed the viruses?

Lines 138-139: Why field-collected mosquitoes were only analyzed at 21 dpi?

Line 144: Please, specify in the serial dilutions were tenfold dilutions.

Line 149: Please, add “/mL” after “ffu”.

Line 150: Please, specify if all tested mosquitoes were full engorged or blood-feed.

Lines 192-194: Please re-write this sentence. Statistics could only support this statement at 21 dpi for Ae. aegypti and with p<0,01 not 0,05.

Line 199: Please, add “tested” after “number of” if it is the case.

Fig. 1: Please modify the scale by indicating the same scale for all the graphics to be more comparable among them at the first glance.

Fig. 1, 3 and 4: Please, edit the y axis to give the proportions in % since the title of the axis is indicated in this way.

Line 222: Please, add “positive” after “number of” if it is the case.

Fig. 2, 3 and 4: Please, change viral particles by ffu/mL in the title of the figure and in the y axis.

Line 253: Delete one “of”.

Fig. 2, 3 and 4: Please, specify which numbers are indicated between brackets to be clearer, since they are different in the different kind of graphics.

Line 284: Please, give write “Eliminate yellow fever epidemics” before the abbreviation EYE if it is the case and provide the reference.

Lines 314-316: Since in the present study dissemination was found for vYFV-247 in Ae. aegypti, it should be pointed out and these statements should be modified to be less categorical.

Lines 320-321: It is true that the insecticide resistance is becoming a problem for vector control, but in case of the mosquito species involved in YFV transmission, the difficulty of insecticide treatment of the breeding sites of these species is also a problem for their control.

Lines 321-324: Please, add a (some) reference(s) for this sentence.

The scientists claim in the abstract that it is improbable that mosquitoes would transmit the vYF-247 vaccination given their findings and the undetectable viremia in vaccinees. All this information, including the reference utilized to claim that the vaccinees have minimal levels of viremia, should be mentioned in the discussion area as well.

Reviewer #2: Recommend major revisions for one reason: vYF-247 needs to be adequately compared to the reference vaccines. Either the experiments need to be optimized to show clear data for the reference vaccines or much of the wording of the paper needs to be changed to reflect the fact that vYF-247 is only successfully compared to wild type YF. Examples of lines that need to be changed: lines 25-28, 42-45, 90-95

Regarding the discussion on viremia of vYF-247 in animal models (line 299-307): To make this point more clear, it would be useful to say something like, “low viremia in these animal models combined with the mosquitoes’ inability to transmit vYF-247 even at the high titers illustrated in this paper indicates that transmission of vYF through a mosquito bite is highly unlikely.” As the paper is currently written, an untrained reader may not understand why the low viremia in animal models is mentioned in the discussion. The way it is stated in the abstract is closer to what I am proposing.

Since viral diversity is mentioned in the discussion, it would also be worthwhile to mention that future studies of vYF-247 should include viral population diversity analysis as another measure showing its non-inferiority to current YF vaccines.

Some of the results paragraphs get a little confusing when percentages and titers are listed one after the other. It may be worthwhile to use semicolons instead of commas to better delineate the data points that are being compared. Example: lines 169-173.

Line 22: Insert the word “a” before “state-of-the-art”

Line 54: change “originated” to “originating”

Line 55: delete “with”. It is redundant after the word “alongside”.

Line 60: This sentence reads awkwardly. Maybe try putting the word “the” before “control” so it reads, “mosquito eradication campaigns of the 1940s and 1950s led to the control of YF…”

Line 74: This would be better described as “epidemic” rather than “pandemic”.

Line 77: mouse tissue was used as well.

Line 180: This sentence reads awkwardly. Try changing it to “infection was only detected at 7 dpi” instead of “there were only infection detected at 7dpi.”

Line 241: swap the words “transmitted” and “successfully”

Line 279: Are the authors saying that the imported cases into China are examples of insufficient vaccination coverage because they should have been vaccinated as travelers to an endemic area? If so, that should be stated.

Line 281: 80% of what?

Line 284: Please spell out the whole name for EYE before using it as an acronym.

Line 286-287: I disagree that the growth of the current YF vaccine in eggs is “the” major blocking point. It would be more realistic to say that it is “a” major blocking point.

Line 310: Check the spelling on “envelope”.

**Summary and General Comments**

Reviewer #1: (No Response)

Reviewer #2: Overall, I think the paper is good. I would suggest major revisions to the abstract and introduction and minor revisions as listed in the sections above.

The authors clearly know the strengths and weaknesses of this paper. The biggest weakness is the lack of data from the reference vaccines. I would suggest that they more clearly state near the beginning of the paper that the reference vaccines could not be compared to the vaccine candidate, vYF-247 (see comments above). Without the reference vaccine data, this data cannot be used by itself to influence the decisions of the WHO for approval of this vaccine candidate. Though the data in comparison to the wild type is clear and helpful.

The strengths of this paper are numerous. The authors clearly state the scope of the issue from the global public health perspective to the much smaller perspective of YF vaccine vector transmission. The study isn’t exactly novel but its significance outweighs that. It fills an important and relevant gap in knowledge about YF vaccine candidates and mosquito transmission. Besides the lack of reference vaccine data, I believe the execution and scholarship of this paper are great. It tells a clear story from point A to point B to point C and solidly connects novel data to the larger public health issues. This analysis is important for researchers and policy makers, alike.

The explanation of how each section of the mosquito correlates to infection, dissemination, and transmission is well done and could be understood by a non-expert, as is the case for much of this paper.

The paragraph from line 317-324 is great. It summarizes the problem well and is easy to understand.

PLOS authors have the option to publish the peer review history of their article (what does this mean?). If published, this will include your full peer review and any attached files.

Reviewer #1: No

Reviewer #2: No
---

## [Decision Letter · Decision Letter 1]

7 Oct 2022

Dear Dr Failloux,

Thank you very much for submitting your manuscript "Absence of transmission of vYF next generation Yellow Fever Vaccine in mosquitoes" for consideration at PLOS Neglected Tropical Diseases. As with all papers reviewed by the journal, your manuscript was reviewed by members of the editorial board and by several independent reviewers. The reviewers appreciated the attention to an important topic. Based on the reviews, we are likely to accept this manuscript for publication, providing that you modify the manuscript according to the review recommendations. 

Sincerely,

Abdallah Samy

Section Editor

Reviewer's Responses to Questions

**Key Review Criteria Required for Acceptance?**

**Methods**

-Are the objectives of the study clearly articulated with a clear testable hypothesis stated?

-Is the study design appropriate to address the stated objectives?

-Is the population clearly described and appropriate for the hypothesis being tested?

-Is the sample size sufficient to ensure adequate power to address the hypothesis being tested?

-Were correct statistical analysis used to support conclusions?

-Are there concerns about ethical or regulatory requirements being met?

Reviewer #1: (No Response)

Reviewer #2: The 17D live attenuated yellow fever vaccine is grown in pathogen-free embryonated eggs. vYF-247 is a vaccine candidate for yellow fever that is produced from 17D adapted to grow in serum-free Vero cells. The authors aim to study vYF-247 infectivity and replication in mosquitoes. They hypothesize that the vaccine candidate will not be transmissible by mosquitoes, especially when compared to a wild type strain of yellow fever virus (YFV). These objectives and hypothesis are clearly stated.

The study design includes testing vYF, two licensed YF 17D-204 vaccines, and wild type YFV in multiple species of mosquito (lab colonies and field-collected). The mosquito experimental techniques and interpretation of results pertaining to infection, dissemination and transmission are in line with the literature. The study design is appropriate for the stated objectives. 

However, the paper states, “The WHO-TRS 978-Annex 5: Recommendations to assure the quality, safety and efficacy of live attenuated yellow fever vaccines recommends to assess mosquito infectivity and dissemination of any new YF vaccine candidates in comparison with a currently acceptable vaccine.” Later in the paragraph the authors note that they test the vaccine strains “when feasible”. This is a good qualifier that implies they weren’t always able to compare the vaccine candidate to the current vaccines. However, I still think it is important that the authors state more clearly that the tests with the current vaccines did not yield significant results. Only the wt vs vaccine candidate tests yielded significant results. Therefore, further comparison to vaccine strains should be listed in the discussion as a future direction.

The populations are clearly described and are appropriate for the hypothesis, which is to assess transmissibility from mosquitoes of vYF-247.

The sample size for the conclusions made is sufficient.

The explanation of the terminology on line 152-156 is very helpful.

The statistical tests utilized (Fisher’s exact test, Mann-Whitney, Kruskal-Wallis) are used correctly.

There are no concerns regarding ethical or regulatory requirements.

For future regulatory requirements, this paper alone should not be used to justify the approval of the vYF-247 vaccine based on mosquito transmissibility. According to WHO requirements, mosquito transmission of any YF vaccine candidate must be compared to current 17D vaccines. This paper does compare vYF to current 17D vaccines but the data is incomplete. Thank you for stating that viremia after vYF-247 vaccination is less than 106.5 ffu/mL; it adds good context to this issue. However, since no pattern or conclusion can be drawn from the 106.5 ffu/mL data, it’s not fair to say that the comparison between vYF-247 and the current 17D vaccines is complete.

**Results**

-Does the analysis presented match the analysis plan?

-Are the results clearly and completely presented?

-Are the figures (Tables, Images) of sufficient quality for clarity?

Reviewer #1: (No Response)

Reviewer #2: I appreciate that the authors changed some of the wording for the comparison between vYF-247 and the vaccine references (e.g., “when feasible”). This makes the analysis presented more clearly represent the analysis plan.

In the experiments that include the YF vaccine references, no real correlation can be seen between vYF-247 and the vaccines, between vYF and the wild type, or between the wild type and the reference vaccines. With this in mind, it is not possible to draw conclusions from the vaccine references (data in the supplemental figures) to the vaccine candidate (data in the main paper).

The authors analyze each mosquito section and mosquito population as intended. They also compare each population tested as intended.

The figures are clear and concise and show all relevant information. The error bars, lines indicating averages, and significance bars are clear. None of the figures are cluttered. The mosquito diagram in figure 1 is particularly helpful to understand the paper. The X and Y axis labels may be a little small but can be easily read on a digital copy. It is easy to distinguish between the colors in each figure.

The results are presented clearly and linearly. They also appear to be completely presented and tell a good story from start to finish.

**Conclusions**

-Are the conclusions supported by the data presented?

-Are the limitations of analysis clearly described?

-Do the authors discuss how these data can be helpful to advance our understanding of the topic under study?

-Is public health relevance addressed?

Reviewer #1: (No Response)

Reviewer #2: The first sentence of the discussion is the main conclusion of this paper and it is supported by the data presented. The conclusions are not overstated and the authors do a good job not to mention comparison to reference vaccines in their statement of the main conclusions. The data collected are easy to understand and the conclusion drawn is clear.

The major limitation of this paper (not being able to compare vYF-247 to the vaccine references) is stated and explained well in the discussion. Their reasoning is understandable and all too relatable. Thank you for changing the wording related to this in the abstract and introduction. However, I believe the wording in the author summary should be changed slightly. On line 44, “but also” is slightly awkward and indicates the reference vaccines were tested to the same degree as vYF-247 and S-79. To make this more clear, it would be beneficial to say something like, “vYF-247 was primarily compared to a clinical isolate and secondarily to reference vaccines, though no significant conclusions could be drawn from the comparisons to the reference vaccines.” It would be useful to have a statement like this in the body of the paper as well so readers do not misinterpret the findings of the paper.

The discussion of viremia in mice and NHPs following inoculation with vYF-247 is extremely relevant for the progression of vYF-247 through future vaccine approval. These data in combination with the mosquito data presented in this paper build a good case that it is very unlikely that vYF-247 can be transmitted by mosquitoes. 

There are many good references to the current literature on YF vaccines from line 305-323. This helps compare vYF-247 to the current YF vaccines and provides good reasoning for the use of vYF-247.

The public health relevance is discussed well. It is good that they are addressing the public health need for YF vaccination as well as the reasoning for developing next-generation YF vaccines. They present real-world goals such as the WHO EYE initiative and use these goals to advocate for their vaccine candidate.

The last paragraph discusses how use of next-generation YF vaccines can be combined with other infection control measures and is a good addition to this paper.

The gap in knowledge that this analysis fills is clear: can this vaccine candidate be transmitted by mosquitoes? It is a relevant and necessary study that will further our understanding of yellow fever vaccines. Lines 33-34 and 312-314 are good statements for real-world objectives.

**Editorial and Data Presentation Modifications?**

Reviewer #1: (No Response)

Reviewer #2: Line 33: Saying that vYF-247 was not transmitted by mosquitoes at “both tested titers” is technically correct but may be slightly misleading. It is important to say what vYF-247 is being compared to in each case. The data the authors have collected in comparison to the wild type at the higher titer is good data! The data for the lower titer is not as good, so saying “both tested titers” here gives the wrong impression.

Line 67: change from “Spillovers were…” to either “Spillovers are…” or “Spillovers can be…”

Line 91-94: As noted above, the authors may want to move this to the discussion section to be used as a future direction.

Apologies if I was unclear about the use of semicolons in the previous revisions!

Lines 172-175: Recommend use of semicolons. “ALPROV was able to become infected at 7 dpi with Stamaril (20%; 8/40); at 14 dpi with Stamaril (7.5%; 3/40), YF-VAX (7.5%; 3/40), and vYF-247 (5%; 2/40); and at 21 dpi with Stamaril (7.5%; 3/40), YF-VAX (7.5%; 3/40), and S-79 (2.5%; 1/40).”

Lines 210-212: for clarity, add semicolons so the section reads like this: “103.2 at 7 dpi, 103.5 at 14 dpi, and 103.3 at 21 dpi in abdomens; 102.6 at 7 dpi, 105.0 at 14 dpi, and 105.5 at 21 dpi in HT; and at 21 dpi, an average of 101.8 particles was detected in saliva.” 

Lines 247-253, 268: may want to add “ffu” after the numbers for clarity on how viral particles are being measured.

Line 296: Could start a new paragraph with “At the outset…” since it is the start of a new topic. May also want to clarify by saying, “At the outset of this study…” to tell the reader that you are no longer talking about clinical trials, but that you are talking about your current mosquito study.

Great clarity on lines 296-304 and lines 308-314.

**Summary and General Comments**

Reviewer #1: (No Response)

Reviewer #2: Overall, I think the paper is good. I would suggest minor revisions to the author summary and other revisions as listed in the sections above.

The authors clearly know the strengths and weaknesses of this paper. The biggest weakness is the lack of data from the reference vaccines. Despite the lack of data on the reference vaccines, the data in comparison to the wild type is clear and helpful.

The strengths of this paper are numerous. The authors clearly state the scope of the issue from the global public health perspective to the much smaller perspective of YF vaccine vector transmission. The study isn’t exactly novel but its significance outweighs that. It fills an important and relevant gap in knowledge about YF vaccine candidates and mosquito transmission. Besides the lack of reference vaccine data, I believe the execution and scholarship of this paper are great. It tells a clear story from point A to point B to point C and solidly connects novel data to the larger public health issues. This analysis is important for researchers and policy makers, alike.

The explanation of how each section of the mosquito correlates to infection, dissemination, and transmission is well done and could be understood by a non-expert, as is the case for much of this paper.

Line 71 is a good statement of one of the problems with controlling YFV.

The paragraph from line 317-324 is great. It summarizes the problem well and is easy to understand.

PLOS authors have the option to publish the peer review history of their article (what does this mean?). If published, this will include your full peer review and any attached files.

Reviewer #1: No

Reviewer #2: No

Figure Files:

Data Requirements:

Reproducibility:

References

---

## [Editor Report · Decision Letter 2]

2 Nov 2022

Dear Dr Failloux,

We are pleased to inform you that your manuscript 'Absence of transmission of vYF next generation Yellow Fever Vaccine in mosquitoes' has been provisionally accepted for publication in PLOS Neglected Tropical Diseases.

Best regards,

Abdallah Samy

Section Editor

---

## [Editor Report · Acceptance letter]

30 Nov 2022

Dear Dr Failloux,

We are delighted to inform you that your manuscript, "Absence of transmission of vYF next generation Yellow Fever Vaccine in mosquitoes," has been formally accepted for publication in PLOS Neglected Tropical Diseases.

Best regards,

Shaden Kamhawi

co-Editor-in-Chief

Paul Brindley

co-Editor-in-Chief
